# Slurry Synthesis and Thin-Film Fabrication Toward Production of Li₂O-B₂O₃-Al₂O₃-Based Multilayer Oxide Solid-State Batteries for Internet of Things Applications

**DOI:** 10.3390/mi16010039

**Published:** 2024-12-30

**Authors:** Jihyun Park, Jongmin Choi, Jihye Seo, Wolil Nam, Soobeom Lee, Seungchan Cho, Kyungchul Park, Geonhyoung An, Beomkyeong Park, Moonhee Choi

**Affiliations:** 1Electronic Convergence Division, Korea Institute of Ceramic Engineering & Technology, 101, Soho-Ro, Jinju 52851, Republic of Korea; pjh0926@pusan.ac.kr (J.P.); hhrqz037@pusan.ac.kr (J.S.); skadnjfdlf@pusan.ac.kr (W.N.); 2School of Materials Science and Engineering, Pusan National University, Busan 46241, Republic of Korea; jm2_choi@krit.re.kr; 3Department of Energy System Engineering, Gyeongsang National University, Jinju 52725, Republic of Korea; soobeomlee@gnu.ac.kr (S.L.); ghan@gnu.ac.kr (G.A.); 4Composites Research Division, Korea Institute of Materials Science, Changwon 51508, Republic of Korea; sccho@kims.re.kr; 5Korea Research Institute for Defense Technology Planning and Advancement, 40, Sadeul-Ro, Jinju 52851, Republic of Korea; kcpark@krit.re.kr

**Keywords:** Internet of Things, microbatteries, microelectronics, oxide-based all-solid-state battery, oxide solid electrolyte

## Abstract

Developing thin-film sheets made of oxide-based solid electrolytes is essential for fabricating surface-mounted ultracompact multilayer oxide solid-state batteries. To this end, solid-electrolyte slurry must be optimized for excellent dispersibility. Although oxide-based solid electrolytes for multilayer structures require sintering, high processing temperatures cause problems such as Li-ion volatilization and reactions with graphite anodes. Thus, low-temperature sinterable oxide-based solid-electrolyte materials should be devised. We successfully developed the conditions for producing thin films from 21 μm thick solid-electrolyte sheets of Li_2_O-B_2_O_3_-Al_2_O_3_, one of the most promising candidates for multilayer solid-state batteries. A comprehensive analysis of the fabricated thin films included X-ray diffraction (XRD) to confirm their amorphous structure, scanning electron microscopy (SEM) for particle morphology, and contact angle measurements to verify surface hydrophilicity. Evaluation of a 32-layer bulk sample of solid-electrolyte sheets revealed an ionic conductivity of 2.33 × 10^−7^ S/cm and charge transfer resistance of 100.1 kΩ at a sintering temperature of 430 °C. Based on these results, cathode and anode active materials will be applied to develop high-energy-density multilayer ceramic batteries with hundreds of layers in future work.

## 1. Introduction

Ultracompact secondary batteries for independent power supply to Internet of Things (IoT) devices are expected to establish a massive market with expansion proportional to the explosive growth in the demand for such devices [1]. As the applications of secondary batteries diversify, global efforts are being devoted to the development of next-generation ultracompact secondary batteries [2,3]. The density of ultracompact sensors and IoT devices in modern electronic devices is rapidly increasing. Moreover, as the device performance improves, ultracompact secondary batteries with high energy density and stable power supply for continuous operation have shown increasing demand. For such applications, conventional organic-electrolyte-based batteries (e.g., lithium-ion batteries) are unsuitable due to their high risk of fire [4,5]. Therefore, high-efficiency secondary batteries with no fire risk must be developed along with surface-mounted devices that can be integrated to supply power to high-density miniaturized devices. As solid-state batteries use stable solid electrolytes instead of the electrolytes in conventional lithium-ion batteries, they offer a substantially higher energy density per unit volume and enable structurally stable thin-film integrated designs [6,7,8]. Oxide-based solid electrolytes are the most promising materials for developing multilayer solid-state batteries, including multilayer ceramic batteries, for internal mounting in IoT devices [9,10,11]. Although oxide-based solid electrolytes enhance the stability and energy density in solid-state batteries for IoT applications, the main challenge is establishing the conditions for uniform thin-film formation using oxide electrolytes [12,13,14,15].

A slurry formulation using ultrafine oxide solid-state battery powders is essential for developing multilayer oxide solid-state batteries. The slurry must achieve excellent dispersibility and stability to produce uniform thin-film solid electrolytes, which are critical for reliable multilayer stacking. Key components of the slurry include dispersants, which prevent agglomeration of oxide particles, and binders, which ensure structural integrity during thin-film formation. Additionally, plasticizers enhance film flexibility, and the solvent system controls viscosity, which directly affects film thickness and uniformity. An improperly formulated slurry can lead to defects, such as inconsistent thickness and reduced ionic conductivity, which significantly degrade the battery’s performance. Based on slurry synthesis, the core technology for forming and stacking oxide-based solid-electrolyte thin-film sheets is essential to create high-performance, reliable multilayer solid-state battery devices.

This study focuses on developing oxide-based thin-film solid electrolytes for multilayer solid-state batteries. The primary objective is to establish low-temperature sintering technology to minimize Li volatilization and enhance ionic conductivity. By optimizing the slurry formulation and evaluating the resulting thin-film properties, we aim to enable the use of ultracompact power supply devices for IoT applications and expand their use to energy storage systems and electric vehicle batteries.

## 2. Materials and Methods

### 2.1. Materials

To produce sample chips of multilayer ceramic batteries, Li_2_O-B_2_O_3_-Al_2_O_3_ (LBA) powder (Posco JK Solid Solution, Yangsan-si, Republic of Korea) with a particle size of 1–2 μm was used as the starting material. Surface treatment of the LBA powder was performed to improve the surface characteristics of the solid-electrolyte thin-film sheet. For surface treatment, the 2-(3,4-Epoxycyclohexyl) ethyltrimethoxysilane (CFS–043, Wuhan, China) coupling agent was used. As solvents for slurry preparation, a mixture of ethanol (Extra Pure Grade, DUCKSAN, Busan, Republic of Korea) and toluene (99.5%, DaeJung Chemicals & Metals, Siheung, Republic of Korea) was used in a 35:55 ratio. To improve dispersibility, BYK–111 (DISPERBYK–111, BYK Additives & Instruments, Wesel, Germany) was added during slurry synthesis. The binder for forming thin-film sheets was an acrylic-based solution (SEN-2600, TTT, Seoul, Republic of Korea), and a plasticizer (Dibutyl phthalate, Daejung Chemicals & Metals, Siheung, Republic of Korea) was added to improve sheet flexibility.

### 2.2. Solid-Electrolyte Surface Treatment

To improve the formability of the thin-film solid-electrolyte sheet, the surface of the solid electrolyte made of LBA powder was treated using a silane coupling agent. The oxide-based solid-electrolyte powder surface was treated with 1 wt% silane coupling agent and underwent stabilization at room temperature for 24 h.

### 2.3. Slurry Preparation for Solid Electrolyte

To prepare the slurry for the solid electrolyte, the surface-treated LBA powder was introduced into an inline mixer chamber (KDM-150, KM Tech, Gimhae-si, Republic of Korea) and dispersed at 150 rpm for 3 h. The acrylic binder, dispersant, plasticizer, and other additives were sequentially added to the initially dispersed LBA slurry. The mixed slurry was further processed using bead milling (NANOSET MILL, DNTEK, Gimpo-si, Republic of Korea) under conditions of 0.1Φ and 4500 rpm for 3 h to obtain the solid-electrolyte slurry (Figure 1).

### 2.4. Oxide-Based Solid-Electrolyte Green Sheet Formation and Green Chip Production

The synthesized solid-electrolyte slurry was used to produce solid-electrolyte thin-film sheets via tape casting (Techgen, Cheongju-si, Republic of Korea). To create thin-film sheets, the gap of the blade coater was set to 150 μm, and a polyethylene terephthalate film was moved at a speed of 0.5 m/min. Drying proceeded at 40–60 °C per section to produce the solid-electrolyte thin-film sheet. The produced solid-electrolyte sheet was stacked to a thickness of 1.5 mm and then compressed under warm isostatic pressure (Techgen, Cheongju-si, Republic of Korea) at 20 MPa. The compressed sheet was cut into 10 × 10 mm pieces using a blade cutter (Techgen) to produce the solid-electrolyte green chips. The green chips were heat-treated at 360 °C for 6 h at a heating rate of 1 °C/min to remove the binder. The binder-free solid-electrolyte green chips were sintered at 430–530 °C under a heating rate of 1 °C/min for 6 h (Figure 2).

### 2.5. Analysis and Evaluation

The viscosity of the synthesized slurry was measured using a viscometer (Brookfield Engineering, Middleboro, MA, USA) to ensure its suitability for making solid-electrolyte sheets. Additionally, the solid content was evaluated using a moisture analyzer (A&D, Tokyo, Japan). The surface characteristics of the solid-electrolyte thin-film sheet made by tape casting were evaluated using a surface roughness tester (Mitutoyo, Kanagawa, Japan) and gloss meter (BYK, Wesel, Germany). To analyze the crystal structure of the sintered solid-electrolyte multilayer chips after layering, X-ray diffraction (D8 Advance A25, Bruker, Billerica, MA, USA) was used with Cu Kα radiation (λ = 1.54 Å) from 20° to 80°. An impedance analysis was conducted using a coin cell jig (Neoscience, Seoul, Republic of Korea) in the frequency range from 7 MHz to 1 Hz to evaluate the electrical properties of the solid-electrolyte chips made under different conditions.

## 3. Results and Discussion

To develop multilayer oxide-based solid-state batteries, a glass material with the LBA composition was selected as the solid electrolyte (Figure 3). The XRD analysis results (Figure 3a) confirmed that it had a broad peak and an amorphous structure. The SEM analysis results (Figure 3b) showed that the particle size was 1 μm, and it appeared to be an asymmetric and non-uniform particle shape.

To improve the electrical properties of the LBA thin-film sheet, the dispersibility of the LBA slurry must be excellent. To enhance dispersibility, strong bonding between the surface of the LBA powder and binder is necessary. Thus, we conducted a wettability test on the LBA powder, and the contact angle was measured at 24.9°, confirming that the LBA solid electrolyte is a hydrophilic material (Figure 4).

The production of hydrophilic LBA solid-electrolyte slurry is the most critical aspect in developing oxide-based multilayer solid-state batteries. To ensure effective bonding between the LBA and binder, the surface of the LBA powder was treated using silane coupling agents (Figure 5). A hydrolysis reaction was first induced by mixing an epoxy-based silane coupling agent with deionized water in a specific ratio. The hydrolyzed silane coupling agent was then mixed with the LBA powder, and a stable surface bonding reaction was induced by allowing the mixture to stabilize at room temperature for 24 h. During this process, hydrogen bonding formed on the powder surface due to the silane coupling agent, enabling the synthesis of a uniformly dispersed slurry when mixed with the binder [16,17,18].

Generally, if the binding between the hydrophilic LBA powder and binder is weak, the powder does not adequately adhere to the binder and remains agglomerated. Then, the flowability of the synthesized slurry decreases, resulting in high viscosity. However, when hydrogen bonds are intentionally created on the surface of the hydrophilic powder using the epoxy-based silane coupling agent, the binding with the binder improves, thereby enhancing the slurry flowability (Figure 6).

Thermal analysis was conducted on the specimens fabricated by stacking and cutting the solid-electrolyte sheet produced from the surface-treated LBA slurry. The analysis results of the compressed and cut specimens revealed that after the onset of solvent evaporation (159.6 °C), a sharp change occurred at 348.4 °C due to binder polymerization and oxidation induced by rising temperature. Subsequently, an exothermic reaction was observed at 457.4 °C due to the crystallization of the oxide solid electrolyte, LBA.

Based on the thermal analysis results of the stacked chips, sintering was performed at 430–550 °C. Comparing the densities of the specimens sintered below the LBA crystallization temperature of 457 °C with those sintered above that crystallization temperature showed comparable sintering densities (Figure 7).

A fracture surface microstructure analysis was also conducted on LBA specimens sintered at various temperatures. For sintering at 430 °C, no crystallization was observed (Figure 8a). In contrast, crystallization was observed for sintering at 450 °C (Figure 8b). Additionally, at the higher temperature of 470 °C, coarse crystal structures were observed (Figure 8c). An elemental analysis of the fracture surface confirmed that the coarse crystals formed at temperatures above 450 °C were LBA crystals (Figure 8d).

Figure 9 shows the impedance analysis results of the LBA multilayer solid-electrolyte specimens sintered at 430–470 °C. The ionic conductivity evaluation from the impedance analysis revealed that the LBA multilayer structure sintered at 430 °C exhibited a resistance of 100.1 kΩ. As the sintering temperature increased, the resistance (R_CT_) increased up to 390.1 kΩ. This increase in resistance was due to crystallization starting at 450 °C (Figure 8b), which generated numerous interfaces, causing a higher resistance to electron transport [19,20,21,22]. It was confirmed that the ion conductivity was 2.33 × 10^−7^ S/cm at 430 °C with a low resistance value, but as the sintering temperature increased, the ion conductivity decreased to 1.4 × 10^−7^ S/cm due to an increase in R_CT_ according to an increase in the crystal structure inside the solid electrolyte.

## 4. Conclusions

This study was aimed to develop ultracompact oxide-based solid-state batteries that can be surface-mounted on printed circuit boards by forming oxide-based solid electrolytes into thin-film sheets and successfully achieve multilayer stacking. To produce high-quality solid-electrolyte green sheets, surface treatment of LBA powder with 1 wt% epoxy-based silane coupling agent followed by 24 h stabilization was the most appropriate approach. An impedance analysis of a 32-layer solid-electrolyte bulk specimen revealed that the optimal temperature for achieving an ionic conductivity of 2.33 × 10^−7^ S/cm and low resistance of 100.1 kΩ was 430 °C, which was a temperature below crystallization onset. The findings of this study will enable further research to develop multilayer ceramic batteries by fabricating cathode and anode active materials as pastes and applying them using screen printing.

## Figures and Tables

**Figure 1 micromachines-16-00039-f001:**
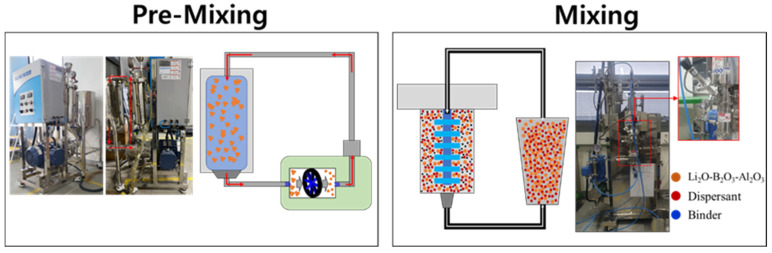
Schematic of dispersion process to prepare slurry for oxide-based solid-electrolyte thick-film sheets.

**Figure 2 micromachines-16-00039-f002:**
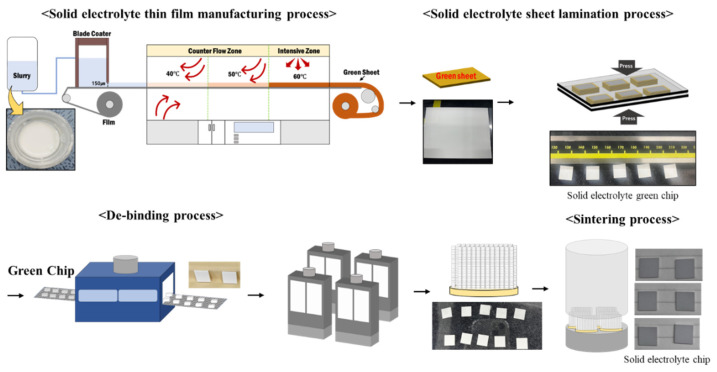
Process flow of oxide-based solid-electrolyte stacking experiment.

**Figure 3 micromachines-16-00039-f003:**
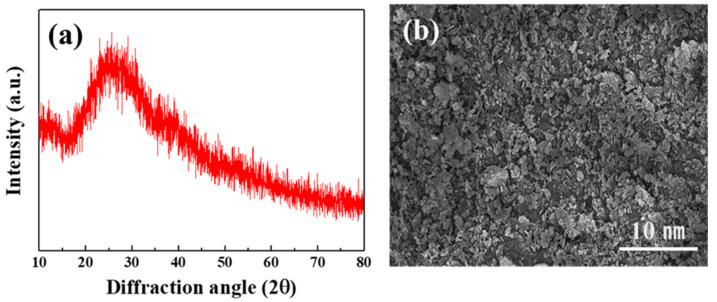
Analysis results of (**a**) crystal structure and (**b**) microstructure of LBA solid electrolyte.

**Figure 4 micromachines-16-00039-f004:**
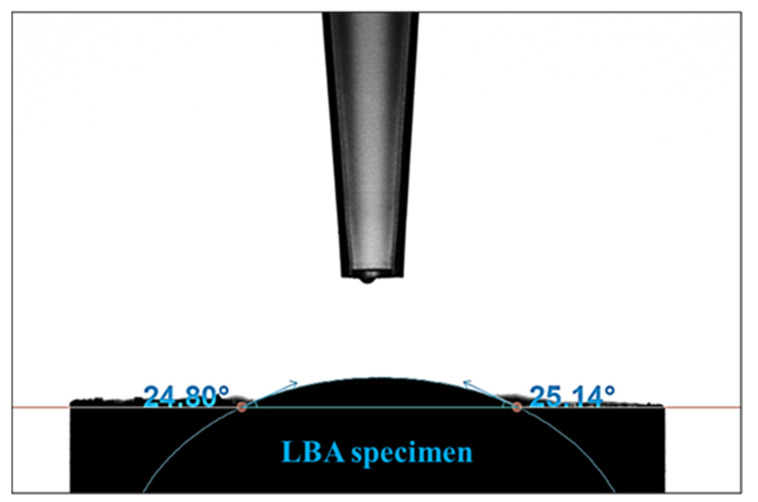
Contact angle analysis of LBA solid-electrolyte material.

**Figure 5 micromachines-16-00039-f005:**
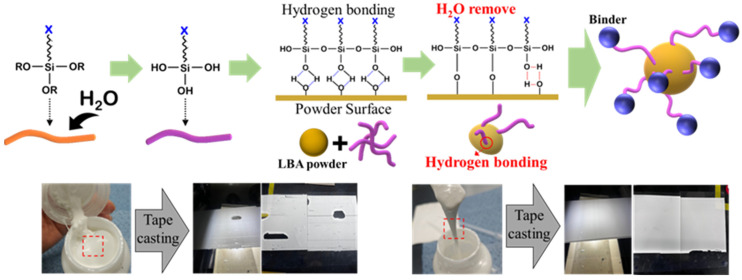
Schematic of LBA surface treatment using silane coupling agents and comparison between slurry and solid-electrolyte sheets produced with and without surface treatment.

**Figure 6 micromachines-16-00039-f006:**
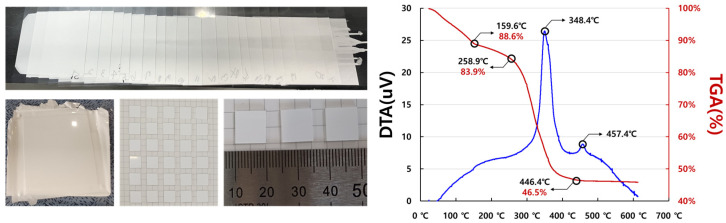
Thermal analysis setup and results for LBA solid-electrolyte thin-film sheet stack produced using surface treatment.

**Figure 7 micromachines-16-00039-f007:**
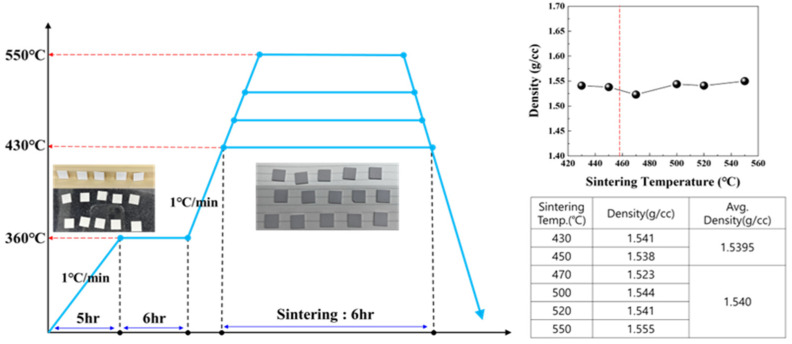
Sintering and density analysis of LBA solid-electrolyte stack produced using surface treatment.

**Figure 8 micromachines-16-00039-f008:**
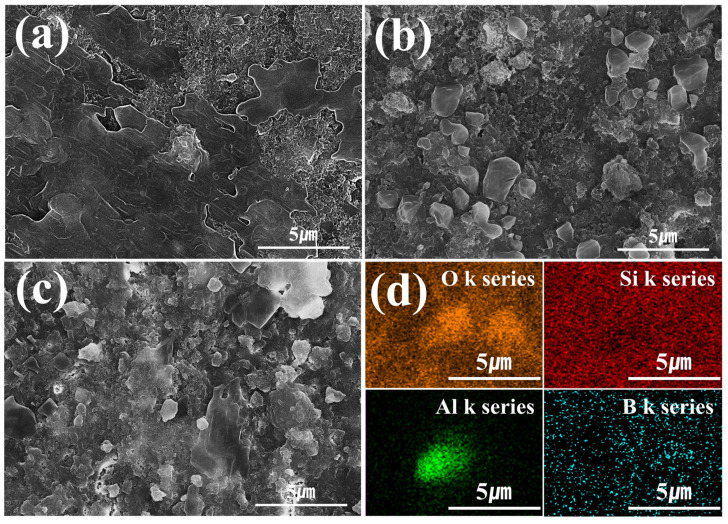
Fracture surface microstructure analysis of LBA solid-electrolyte stack sintered at temperatures of (**a**) 430, (**b**) 450, (**c**) 470 and energy-dispersive X-ray spectroscopy analysis of LBA solid-electrolyte stack sintered at temperatures of (**d**) 470 °C.

**Figure 9 micromachines-16-00039-f009:**
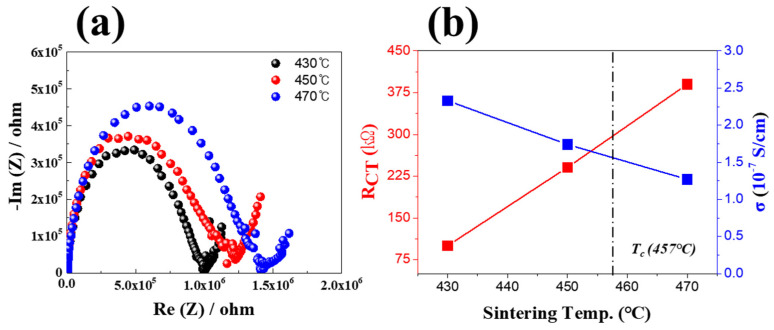
Impedance analysis results of LBA solid−electrolyte multilayer stacks sintered at different temperatures. (**a**) Impedance analysis results and (**b**) impedance and ionic conductivity characteristics.

## Data Availability

Data Availability Statements are available in section.

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
