# Peer review of "Slurry Synthesis and Thin-Film Fabrication Toward Production of Li₂O-B₂O₃-Al₂O₃-Based Multilayer Oxide Solid-State Batteries for Internet of Things Applications"

_micromachines, 2024, doi:10.3390/mi16010039_

Round 1

Reviewer 1 Report

Comments and Suggestions for Authors

Review comments

The manuscript presents a comprehensive study on the development of Li₂O-B₂O₃-Al₂O₃-based multilayer oxide solid-state batteries for Internet of Things (IoT) applications. The authors have successfully optimized the slurry synthesis for oxide-based solid electrolytes, achieving excellent dispersibility and low-temperature sintering, which is crucial for preventing lithium ion volatilization and reactions with graphite anodes. The innovation in this work is the significant enhancement of ionic conductivity and the development of thin-film solid-electrolyte sheets with promising properties for multilayer solid-state batteries. This study not only provides a new approach to fabricating ultracompact power supply devices for surface-mounted IoT applications but also deepens the understanding of the relationship between material processing and battery performance.

I believe that with a few minor revisions, this manuscript would be a valuable contribution to the journal.

1.     The authors could further discuss how their findings compare with other results in the existing literature, particularly regarding the development and application of oxide-based solid-state batteries. This will clearly articulate the novelty and unique contributions of this study.

2.     Given the intended applications in IoT devices, it is important to understand the mechanical and thermal stability of the developed solid-state batteries. The authors are encouraged to include data on the mechanical strength, flexibility, and thermal stability of the thin-film solid-electrolyte sheets.

Reviewer 2 Report

Comments and Suggestions for Authors

Dear Sir,

the authors are developing a low-temperature sintering technology for the synthesis of oxide-based thin-film solid electrolytes to implement multilayer solid-state batteries.

The paper is interesting and well prepared however it can be improved in some areas as follows:

In the abstract

Some grammatical error should be corrected and there are some typos in the manuscript for example:

-       Please state exactly the performed analysis regarding the characterization of the obtained films such as XRD, SEM etc.

-       This sentence may be confusing “we successfully developed the conditions for producing 21 μm thick thin-film” I suggest you can use “we successfully developed the conditions for producing thin-film of 21 μm thick “

-       keywords should be arranged alphabetically and number of  keywords can  be decreased 5 is reasonable.

Introduction section:

Some grammatical error should be corrected and there is some typos in the manuscript for example:

-       Line 48 promising materials

-       More explanation for the slurry technique should be provided for the reader.

-       The last part of this section should be shorten and focus only the main aim of the research.

In the other sections:

-       Could the author give more information about the XPS analysis in particular the ratio of the metals in the film?

-       EDX analyses could give further information on the chemical composition of the prepared samples

-       Is there any possible interference between the thin film and the protective layer so did authors study that practically?

-       Authors should make a comparison of their results with the previously articles that similar work to have clear understanding of their research impact.

-       How many replicate did you perform as reproducibility should be stated?

Regards

Comments on the Quality of English Language

The quality of English language is good.

Round 2

Reviewer 2 Report

Comments and Suggestions for Authors

Dear Author,

thanks for your efforts and for the detailed explanation for the suggestions and the recommended references. I wish you could perform more characterization in the your future research.